# Assessment of the Effects of Triticonazole on Soil and Human Health

**DOI:** 10.3390/molecules27196554

**Published:** 2022-10-03

**Authors:** Diana Larisa Roman, Denisa Ioana Voiculescu, Mariana Adina Matica, Victor Baerle, Marioara Nicoleta Filimon, Vasile Ostafe, Adriana Isvoran

**Affiliations:** 1Department of Biology-Chemistry, Faculty of Chemistry, Biology, Geography, West University of Timisoara, 16 Pestalozzi, 300115 Timisoara, Romania; 2Advanced Environmental Research Laboratories (AERL), 4 Oituz, 300086 Timisoara, Romania; 3Department of Chemistry-Biology, Institute for Advanced Environmental Research (ICAM), West University of Timisoara, Oituz 4C, 300086 Timisoara, Romania

**Keywords:** stereospecificity, ecotoxicity, human toxicity

## Abstract

Triticonazole is a fungicide used to control diseases in numerous plants. The commercial product is a racemate containing (R)- and (S)-triticonazole and its residues have been found in vegetables, fruits, and drinking water. This study considered the effects of triticonazole on soil microorganisms and enzymes and human health by taking into account the enantiomeric structure when applicable. An experimental method was applied for assessing the effects of triticonazole on soil microorganisms and enzymes, and the effects of the stereoisomers on soil enzymes and human health were assessed using a computational approach. There were decreases in dehydrogenase and phosphatase activities and an increase in urease activity when barley and wheat seeds treated with various doses of triticonazole were sown in chernozem soil. At least 21 days were necessary for the enzymes to recover the activities. This was consistent with the diminution of the total number of soil microorganisms in the 14 days after sowing. Both stereoisomers were able to bind to human plasma proteins and were potentially inhibitors of human cytochromes, revealing cardiotoxicity and low endocrine disruption potential. As distinct effects, (R)-TTZ caused skin sensitization, carcinogenicity, and respiratory toxicity. There were no significant differences in the interaction energies of the stereoisomers and soil enzymes, but (S)-TTZ exposed higher interaction energies with plasma proteins and human cytochromes.

## 1. Introduction

Modern agriculture is still vastly reliant on pesticides to assure high yields and product quality, the azoles being the fungicide class used commonly to control fungal plant diseases, accounting for 20–25% of the global market [1]. Triticonazole (TTZ), IUPAC (International Union of Pure and Applied Chemistry) name 5-[(4-chlorophenyl)methylidene]-2,2-dimethyl-1-(1,2,4-triazol-1-ylmethyl)cyclopentan-1-ol, belongs to this class being a triazole fungicide widely used as a seed treatment for the control of common soil and seed diseases in cereals, turfgrass, and ornamental plants [2], for example, the numerous foliar pathogens [3], legume, and fruit tree diseases [4]. In the case of seed application, TTZ prevents the development of fungi mycelium both on or inside seeds and in soil. Similar to other triazole fungicides, its mode of action is based on the inhibition of demethylation in the sterol biosynthesis pathway in most fungi [3]. TTZ is a racemic compound, i.e., it contains equal amounts of the two stereoisomers, (R)- and (S)-triticonazole. The two stereoisomers reveal distinct fungicidal activity, (R)-triticonazole being at least three times more potent than (S)-triticonazole [5,6]. Furthermore, (S)-TTZ has higher acute toxicity to non-target organisms (both aquatic and terrestrial) than (R)-TTZ and the racemic mixture [7].

Data from the literature show that triazole fungicides are quite persistent in soil; they decrease the soil enzyme activities and affect the population of microorganisms found in soil and aquatic environments [8,9]. TTZ has been identified in agricultural soils from China with a median value of 1.29 ng per g of soil [10] and proved to be very persistent in soil with a 90% degradation period of 329–803 days [11]; however, we were not able to identify consistent information regarding the TTZ effects on the activities of soil enzymes and the soil microbial population. A study by Charnay et al. 2000 [12] revealed that the microbial C biomass decreased by 8.5% after 70 days of aerobic incubation under laboratory conditions in samples of loamy clay soil from the northern region of France treated with TTZ. Another study revealed no significant adverse effects of TTZ on microorganisms involved in nitrogen and carbon mineralization [11]. Furthermore, no information was available regarding the effects of the TTZ stereoisomers on soil enzymes. This emphasizes the need to assess the effects of this fungicide and its isomers on the soil environment.

The connections between human health and the environment are complex, with environmental pollution mediating the transmission of diseases and posing a number of health risks to humans [13]. Environmental contamination from pesticides and their degradation products present in the air, water, soil, and food, together with their occupational use and their production sites near crops, are the pathways of exposure of humans to pesticide residues [14]. Regarding TTZ, humans are exposed to its enantiomers, the pesticide residues being detected in agricultural products [15], and drinking water [16]. Data in the literature reveal several toxicological effects of TTZ on mammals. This fungicide has shown inhibition of cytochrome P450 enzymes (CYP), especially CYP51, and strong androgen receptor antagonism [17]. A computational study showed that TTZ could produce human toxicity as it was able to penetrate the blood–brain barrier to inhibit CYP2C9 and CYP2C19 and cause skin sensitization, carcinogenicity, and moderate endocrine disrupting effects [18]. Field studies demonstrated that (S)-triticonazole was more persistent than (R)-triticonazole in fruits and vegetables, the two stereoisomers registering distinct degradation rates; (R)-TTZ was preferentially degraded in Chinese cabbage, cucumber, tomato, and spinach [5] and (S)-TTZ registered higher degradation rates in pears, peaches, and jujube [19]. To the best of our knowledge, no information is available regarding the biological effects of TTZ stereoisomers on human health. Furthermore, quite distinct biological effects on humans of the stereoisomers of difenoconazole, another widely used triazole fungicide, have been observed [20]. This underlines the need to assess the possible human health effects of TTZ stereoisomers.

Taking into account the lack of information regarding the effects of TTZ on both soil environment and humans, the aim of this study was to assess the effects of TTZ on the activity of enzymes considered soil quality indicators (dehydrogenase, urease, phosphatase) on soil microbial communities and human health and consider the TTZ enantiomeric structure when appropriate. For this purpose, a combination of experimental and computational approaches was considered. The experimental approach was based on determining the activities of soil dehydrogenases, ureases, and phosphatases and assessing the population of the communities of microorganisms for the soil samples containing barley and wheat seeds treated with distinct doses of TTZ. The computational approach was based on predicting the absorption, distribution, metabolism, excretion, and toxicity (ADMET) profile of the TTZ enantiomers and using molecular docking for assessing the interactions of the TTZ enantiomers with the investigated soil enzymes, human cytochromes involved in the metabolism of xenobiotics, and plasma proteins, respectively. Usually, the software that allows predictions of ADMET profiles is used for screening drug candidates, but it proved to also be useful for assessing the effects of various types of chemicals on both humans and the environment: drug-related molecules [21,22,23,24,25], pesticides [18,20,26,27], sweeteners [28], phthalates [29], food additives [26], and industrial chemicals [30]. These data reveal that these methods are applicable for assessing the biological effects of chemicals.

## 2. Materials and Methods

### 2.1. Fungicides

The experiments were performed using a product marketed locally under the trade name “Premis”, containing 25 g/L triticonazole. The product is applied for seed treatment before sowing. Triticonazole with a molecular weight of 317.8 g/mol and XlogP = 3.1 has one chiral center and contains two stereoisomers, (R)- and (S)-triticonazole. The technical material is a racemate, i.e., it comprises equal amounts of the two stereoisomers [31]. The SMILES (Simplified Molecular Input Line Entry System) formulas of the two stereoisomers of TTZ were extracted from the ZINC database [32], and the three-dimensional structures of the stereoisomers were built using Chimera software [33] and are presented in Figure 1.

### 2.2. Soil Sampling

The soil (chernozem) samples were collected in March 2022 from a field located close to Timisoara city, Romania, (45°45′14.54″ N, 21°18′16.66″ E), from an area with no known history of pesticide use. The soil samples were taken from the top layer of soil (maximum 20 cm) from 5 distinct locations in quantities of 20 kg each. The soil was ground, sieved (2 mm), and spooned by random sampling to obtain sub-samples that were immediately processed. Experiments were performed on control soil samples (soil containing wheat or barley seeds untreated with TTZ) and soil samples sown with seeds of wheat and, respectively, barley treated with TTZ, grouped into experimental variants, as presented in Table 1. Wheat and barley were chosen for this study because it is known that in Europe, azoles are widely used for treating cereal crops, especially wheat and barley [1]. The doses that were used in the present study correspond to the recommended minimum and maximum doses by the producer for wheat and barley seeds. An intermediary dose was also considered. The application of the fungicide was performed by spraying on the seeds of wheat and barley before sowing for each experimental variant. Wheat and barley seeds treated with TTZ were mixed with the soil samples.

### 2.3. Enzymatic Activity Analyses

The activities of the following enzymes were investigated in this study: dehydrogenase (EC 1.1.1.1), urease (EC 3.5.1.5), and phosphatase (EC 3.1.3.2) using the spectrophotometric method. When determining dehydrogenase activity (DA), the method described by Schinner and coworkers was used [34]. In the case of the urease activity (UA), the method proposed by Alef and Nannipieri was considered, [35] and when investigating the phosphatase activity (PhA), the method exposed by Dick was used [36]. Details regarding the implementation of these methods were also presented in one of our previous studies regarding the assessment of the effects of the herbicide S-metolachlor on the activity of soil enzymes [37]. Determination of each enzymatic activity was performed every 7 days during a 28-day incubation period under laboratory conditions of the soil sown with wheat and barley seeds treated with TTZ. All measurements of enzymatic activities were performed using a T90 UV/Vis spectrophotometer (PG Instruments, England). For each soil sampling, the investigation was carried out in triplicate for each experimental variant by the same researcher on the same day.

### 2.4. Microbiological Analysis

The count of microorganisms (bacteria and fungi) in soil samples was carried out using serial dilution and the spread plate method [38] as follows: for each sample, 1 g of soil was weighed and placed in 9 mL of sterile distilled water and shacked for 10 min at 250 rpm at room temperature (RT). After removing the test tubes from the shaker, while the soil particles were suspended in water, 1 mL of soil suspension was added to 9 mL of sterile distilled water. This gave a dilution of 10-2, which was used for the fungi counting, whereas for the bacterial counting, the steps described above were repeated once more until a dilution of 10-3 was reached. For the bacterial isolation, Plate Count Agar (agar 15 g/L, glucose 1 g/L, enzymatic digest of casein 5 g/L, yeast extract 2.5 g/L) was used, and for fungi cultivation, Peptone Yeast Agar (soya peptone 10 g/L, yeast extract 5 g/L, glucose 40 g/L, streptomycin sulfate 0.03 g/L, chloramphenicol 0.05 g/L, agar 15 g/L) was used. A volume of 0.1 mL inoculum was spread on the surface of the agar media and the plates were incubated at 28 °C. For the bacteria enumeration, the plates were counted after 24 h of incubation, whereas for the fungi count, the plates were analyzed after 48 h using an automated colony counter (SCAN 300 Interscience). The fungi and mold colonies from the plates were counted separately. All samples were performed in triplicate and the colony forming units (CFU) were calculated and expressed as CFU/g soil samples.

### 2.5. Prediction of the Effects on Human Health of the Stereoisomers of Triticonazole

The ADMETlab2.0 online tool (https://admetmesh.scbdd.com/service/screening/index, accessed on 12 April 2022) was considered for obtaining quantitative predictions regarding the human health hazards of TTZ. This tool was used as the entry data for the SMILES formulas of the investigated molecules and it output the predictive values for some properties and biological activities and the prediction probabilities for other activities to be manifested by the investigated chemical, respectively. Several of the ADMET properties were predicted using regression models (absorption, plasma protein binding, fraction unbound in plasma, volume of distribution, clearance, environmental toxicity) and other properties and toxicological endpoints were predicted based on classification models (P-glycoprotein inhibitor and substrate, blood–brain barrier penetration, inhibitor/substrate of human cytochromes involved in the metabolism of chemicals, skin sensitization, hepatotoxicity, cardiotoxicity, nephrotoxicity, mutagenicity, carcinogenicity, etc.). The classification models had a minimum accuracy of 0.80 and for most of the regression models R^2^ > 0.72 [39,40]. ADMETlab2.0 was chosen for predicting the human health hazards of TTZ as it proved to be more adequate for investigating the ADMET profiles of the chiral molecules [22].

As reported in specific literature, since triazole fungicides are able to produce endocrine disruption in mice [17,41,42], the ENDOCRINE DISRUPTOME [43] computational facility was used to predict the possible effects of TTZ stereoisomers on the endocrine system of humans. ENDOCRINE DISRUPTOME is based on the molecular docking that is implemented for 14 human nuclear receptors; the following best structures were selected from the Protein Data Bank (PDB) [44]: androgen receptor (AR); estrogen receptors α and β (ER α and ER β); progesterone receptor (PR); glucocorticoid receptor (GR); liver X receptors α and β (LXR α and LXR β); mineralocorticoid receptor (MR); peroxisome proliferator-activated receptors α (PPAR α), β/δ (PPAR β), and γ (PPAR γ); retinoid X receptor α (RXR α); and thyroid receptors α (TR α) and β (TR β). The chemical compound was docked to these structures and a docking score of the compound docking on every receptor structure was computed. The docking scores and the validation experiments were used to compute the probability thresholds that were translated into the sensitivity parameters (SE). These parameters were further used to classify the binding of the investigated compound to the considered nuclear receptors: SE < 0.25 corresponded to a high probability of binding, 0.25 < SE < 0.50 corresponded to a mean probability of binding, 0.50 < SE < 0.75 revealed a moderate probability of binding, and SE > 0.75 corresponded to a low probability of binding [43].

### 2.6. Molecular Docking Study

In order to evaluate the stereospecificity of the effects of the isomers of TTZ on the enzymes used as the soil quality indicators, the human cytochromes involved in the metabolism of xenobiotics, and the human plasma protein binding proteins, the molecular docking approach was considered. The two TTZ stereoisomers were docked with several structures of enzymes occurring in soil intracellular in microbial cells, such as dehydrogenases [45], extracellular secreted by microbes, such as phosphatases [46], or both intracellular and extracellular, such as ureases [47]. Similarly, the two TTZ enantiomers were docked to structures of human plasma proteins (alpha 1-acid glycoprotein, and human serum albumin) and human cytochromes (CYPs): CYP1A2, CYP2C9, CYP2C19, CYP2D6, CYP3A4. The structures of the TTZ stereoisomers were extracted from the ZINC database [32] and prepared for docking using Chimera software [33]. The structures of the soil enzymes and human proteins were selected from the Protein Data Bank (PDB) [44]. The structural files that were selected for molecular docking were revealed and are briefly characterized in Appendix A [48,49,50,51,52,53,54,55,56,57,58,59,60,61]. These structural files were chosen so as to belong to complexes made by proteins with ligands and to have the best resolution among the existent crystallographic structures. Analysis of the crystallographic structures of the different chains in the multimeric proteins revealed that the ligands did not always bind to the same amino acids in every chain of the same protein and that some chains did not contain ligands. Consequently, to take into account the flexibility of the active sites of the investigated proteins, molecular docking was implemented for each protein chain in the cases of multimeric crystal structures.

### 2.7. Statistical Analysis

Data were analyzed using variance analysis, the software employed was MIMITAB 14 (Minitab LLC, State College, PA, USA). All data are presented as mean values with standard deviation (X ± SD). The significant difference in the variables was tested using the Mann-Whitney test at a 0.05 level of probability.

## 3. Results

### 3.1. Effects of Triticonazole on the Activity of Enzymes Found in Soil

The activities of soil dehydrogenases, ureases, and phosphatases were recorded for 28 days for the soil samples containing barely and wheat seeds treated with distinct doses of triticonazole and observed every 7 days after treatment, as presented in Figure 2, Figure 3 and Figure 4.

In both cases where the soils were sown with barley and wheat seeds not treated with TTZ (negative control samples), DHA strongly decreased throughout the incubation period, PhA decreased in the first 14 days, and UA increased in the first 7 days and then decreased further. These fluctuations are probably due to the fact that the soil samples were maintained under laboratory conditions. Furthermore, the values of DHA for barley and wheat seeds were quite distinct. This can be explained by the fact that the DHA at the level of the soil sown with wheat and barley seeds was determined by the composition and complexity of the community of microorganisms existing around the seeds. The different morphological aspects (shape, size) of wheat and barley seeds influenced the amount of triticonazole absorbed and further dissipated in the soil around the seeds. Consequently, a distinct amount of fungicide was released around the two types of seeds, differently affecting the community of the soil microorganisms and implicitly the enzyme activity, especially the DHA. Moreover, the differences in the microbial community, thus the soil enzyme activity, could have also resulted from the two types of roots with distinct metabolic activities resulting in different chemical compositions and the amounts of litter produced.

The variations in the soil enzyme activity for each of the experimental samples containing barely or wheat seeds treated with various doses of TTZ usually followed the same temporal behavior, but there were distinct values for this activity during the incubation time when the samples were compared to each other and the control variants.

In the case of the soil sown with seeds treated with TTZ, after 7 and 14 days of incubation, there were significant decreases (*p* < 0.05) in the DHA and PhA and a significant increase in the UA (*p* < 0.05) compared with the negative control, regardless of the dose of TTZ. After 21 days, the DHA was still lower than in the control experimental variants. In the case of the PhA, starting at 21 days, it increased but was not significantly different from that obtained in the control experiments. The UA still increased after 14 days and decreased starting at 21 days compared to the UA in the control experiments, but these differences were not significant. We must also mention that there were no significant differences in the enzyme activities between one TTZ dose and another.

These outcomes reveal that in the soil samples containing barley and wheat seeds treated with TTZ, the DHA, PhA, and UA needed at least 21 days to recover and were always significantly lower for the soil samples containing barley seeds treated with TTZ compared to the soil samples containing wheat seeds treated with TTZ. This underlines the influence of the type of crop on soil activity. Also, the differences that appeared between the activities of the soil enzymes in the soil samples analyzed may be due to the ability of communities of soil microorganisms to ensure the restoration of the initial ecological balance. The effects of the seed treatments with various doses of TTZ did not appear to be dose dependent.

The results obtained in these experiments are in good agreement with the published data regarding the effects of other triazole fungicides on the activity of enzymes found in soil. Several published papers showed that difenoconazole applied in various concentrations produced a decrease in the DHA, PhA, and UA [49,50]. Myclobutanil [62,63], paclobutrazol [64], and triadimefon [61] applications also produced decreases in the DHA. The propioconazole application conducted caused a decrease in the PhA and UA [65]. The presence of tebuconazole in the soil samples led to a decrease in the DHA, PhA, and UA [66,67,68,69,70]. Triazole fungicides were also shown to affect the activity of other enzymes found in soil: chitinases [71], arylsulfatase, β-glucosidase, invertase, and catalase [66,68,70]. The decreases in the activities of these enzymes have been observed in various types of soil over different periods of time under laboratory and/or field conditions and were proven to be dose- and time-dependent.

### 3.2. Effects of Triticonazole on the Soil Microorganisms

The effects of the treatment of the barley and wheat seeds with the TTZ on the communities of soil microorganisms were recorded every 7 days for 28 days and are presented in Figure 5 and Figure 6.

Data presented in Figure 5 illustrate that TTZ doses applied to the barley and wheat seeds did not significantly affect the soil population of bacteria during the 28 days of incubation. In addition, the registered bacterial population did not significantly differ in the samples containing wheat and barley seeds treated with TTZ. This outcome is in good agreement with the results presented in another study, which showed the insignificant adverse effects of this fungicide on microorganisms involved in nitrogen and carbon mineralization [11].

Statistical analysis of the evolution of the entire soil microorganism community (bacteria and fungi) showed that there was a significant decrease (*p* < 0.05) in the number of microorganisms only in the case of the samples containing barley seeds treated with the maximum dose of TTZ after 14 and 21 days of incubation. This was in strong correlation with the registered activities for the investigated soil enzymes that revealed that at least 21 days were necessary for recovery and the enzyme activities were always significantly lower for the soil samples containing barley seeds treated with TTZ. Establishing the effect of the TTZ on non-target microorganisms is important because soil microorganisms are necessary for maintaining healthy soil.

### 3.3. Molecular Docking Study Regarding the Interactions of the Enantiomers of Triticonazole with Soil Enzymes

The molecular docking approach was considered for assessing the interactions of the two stereoisomers of TTZ (R- and S-TTZ) with the enzymes found in soil: dehydrogenases, phosphatases, and urease. Several types of dehydrogenases found in soil microorganisms, three types of phosphatases, and one urease secreted by soil microorganisms were considered for molecular docking (Table 2). The results obtained by the molecular docking study on the binding modes corresponding to the highest interaction energies recorded for the binding of the (R)- and (S)-TTZ to the *Clostridium beijerinckii* NADP-dependent alcohol dehydrogenase and, respectively, to the aldehyde dehydrogenase from *Bacillus cereus* are presented in Figure 7.

Figure 7a shows that the binding poses corresponding to the higher interaction energies for the (R)- and (S)-TTZ to the *Clostridium beijerinckii* NADP-dependent alcohol dehydrogenase were quite similar and corresponded to the binding region of the NADPH. Figure 7b emphasizes that in the case of aldehyde dehydrogenase from *Bacillus cereus*, only the (S)-TTZ was able to bind to the catalytic site of the enzyme.

The two TTZ enantiomers were able to bind to the catalytic sites of several enzymes, as presented in Table 2. For comparison purposes, Table 2 also contains the interacting energies registered for the ligand that was present in the crystallographic structure for each protein, when available. The highest energies were recorded for the interactions of the TTZ stereoisomers with *Clostridium beijerinckii* NADP-dependent alcohol dehydrogenase and with aldehyde dehydrogenase from *Bacillus cereus*. In the case of the *Bacillus stearothermophilus* phosphatase PhoE (PDB ID 1H2F), the adenosine monophosphate was used in the molecular docking study being as it is known that this molecule is one of the substrates of this enzyme [72]. In the cases of the acid phosphatase from *Aspergillus niger* (PDB ID 1QFX) and the alkaline phosphatase from *Bacillus subtilis* (PDB ID 2YEQ), there were no ligands bound in their active sites. Data from the literature show that phytase (myo-inositol hexakisphosphate) is the substrate for acid phosphatase from *Aspergillus niger* [73]. The molecular docking study of the interaction of phytase with this enzyme revealed that phytase was able to bind to the active site of the enzyme, with an interaction energy of −10.22 kcal/mol. In addition, 4-nitrophenyl phosphate (pNPP) is known as being the substrate for the alkaline phosphatase from *Bacillus subtilis* [54], but the molecular docking study revealed that pNNP was not able to bind to the active site of the enzyme. In the case of *Bacillus pasteurii* urease (4AC7), the molecular docking study did not produce binding poses for citrate (the ligand present in the crystallographic structure) corresponding to the active site of the enzyme. When docking the urea (the specific substrate) to the *Bacillus pasteurii* urease, the molecular docking emphasized that urea was able to bind to the catalytic site (ΔG = −8.89 kcal/mol). This can be explained by the fact that the binding of citrate to urease is based on a concentration-dependent activation/inhibition mechanism. At low concentrations, citrate binds to an activation site that has not yet been characterized, whereas at high concentrations such as those used in the determination of the urease–citrate complex, citrate binds to the enzyme active site and has an inhibitory effect [62].

**Table 2 molecules-27-06554-t002:** Interaction energies (ΔG) between the stereoisomers of triticonazole (TTZ) and soil enzymes for the best docking poses corresponding to the active sites of enzymes. This table also contains the interacting energies obtained for the docking of the ligands that were present in the crystallographic structures of the investigated enzymes: NADPH—dihydro-nicotinamide-adenine-dinucleotide phosphate, FBP—1,6-di-O-phosphono-beta-D-fructofuranose, NAI—1,4-dihydronicotinamide adenine dinucleotide, ISE- inositol, AMP—adenosine monophosphate, pNPP—4-nitrophenyl phosphate.

Soil Enzyme	PDB ID	Ligand in Crystallographic Structure			(R)-TTZ		(S)-TTZ	
		Ligand	ΔG (kcal/mol)	Mean Value of ΔG (kcal/mol)	ΔG (kcal/mol)	Mean Value of ΔG (kcal/mol)	ΔG (kcal/mol)	Mean Value of ΔG (kcal/mol)
Dehydrogenases	1KEV chain A	NADPH	−16.70	−18.67	−10.06	−10.36	−10.04	−10.22
	1KEV chain B	NADPH	−16.84		−10.60		−10.09	
	1KEV chain C	NADPH	−20.98		−9.99		−10.38	
	1KEV chain D	NADPH	−20.16		−10.81		−10.38	
	3AUT chain A	NAI *	−10.32	−10.42	−7.21	−7.31	−7.70	−7.58
	3AUT chain B	NAI	−10.51		−7.41		−7.46	
	3NT5 chain A	ISE	−6.51	−6.50	−7.72	−7.29	−7.94	−7.52
	3NT5 chain B	ISE	−6.59		−6.86		−7.09	
	5GTL chain A	NADPH	−14.88	−15.78	it did not bind in the active site	−9.37	−9.16	−9.46
	5GTL chain B	NADPH	−17.11		−9.46		−9.94	
	5GTL chain C	NADPH	−13.54		−9.72		−9.16	
	5GTL chain D	NADPH	−14.57		−8.93		−9.58	
Phophatases	1H2F	AMP *	−8.89	−8.89	−8.19	−8.19	−8.37	−8.37
	1QFX chain A	phytate *	−10.22	−10.94	it did not bind in the active site			
	1QFX chain B		−11.65					
	2YEQ chain A	pNPP *	it did not bind in the active site					
	2YEQ chain B							
Urease	4AC7 chain C	urea *	−6.47	−6.47	it did not bind in the active site			

* These molecules were not found in the crystallographic structure but they are known as being the specific substrates for these enzymes [49,54,72,73].

Despite the experimental data revealing the decrease in the PhA and UA in the presence of TTZ, neither (R)-TTZ nor (S)-TTZ could bind to the active sites of the alkaline phosphatase from *Bacillus subtilis* (2YEQ), the acid phosphatase from *Aspergillus niger* (1QFX) and the *Bacillus pasteurii* urease (4AC7). These situations may be due to the fact that TTZ can affect other microorganisms that are present in soil or the specific structures and properties of the catalytic sites of these enzymes. Some structures were in closed conformations and the lack of flexibility of both the protein and ligand in the molecular docking study did not conduct enzyme–ligand interactions corresponding to the binding sites. The active site of the alkaline phosphatase from *Bacillus subtilis* is hydrophilic and its closed conformation is mentioned in the description of the structure, there being a C-terminal helix lying over the active site and controlling the access to this site [54] (Appendix A). The acid phosphatase from *Aspergillus niger* (1QFX) had a highly charged and consequently hydrophilic active site (Appendix A) and this may explain why the hydrophobic TTZ enantiomers could bind. In the case of *Bacillus pasteurii* urease, its catalytic site was narrow and highly hydrophilic and there was a considerable rigidity of the protein scaffold around the active site, the catalysis being complemented by hydrogen bonds stabilizing the binding of the substrate and by the movement of a flexible flap (a 300–350 helix–loop–helix region) that changed the active site into a closed conformation [62] (Appendix A).

The outcomes obtained using PLIP [74] software regarding the identified interactions between the two TTZ stereoisomers and the enzymes considered in this study are presented in Appendix A. For comparison purposes, PLIP software was also used for identifying the interactions appearing in the complexes of these enzymes and their ligands that were present in the crystallographic structures or in the complexes obtained by docking some of these enzymes with their specific substrates (as explained previously). These interactions are also shown in Appendix A. The noncovalent interactions of the TTZ stereoisomers with the investigated proteins were hydrophobic and hydrogen bonds. The spectra of the interactions made with the residues of the soil enzymes were quite different from one isomer to another, emphasizing the specificity of the binding of each stereoisomer to these enzymes and in good correlation with the distinct interaction energies for the two stereoisomers.

### 3.4. Predictions of the ADMET Profiles of the Enantiomers of Triticonazole

The outcomes of ADMETLab2.0 [39,40] regarding the absorption, distribution, and excretion of the stereoisomers of TTZ are shown in Table 3, those regarding the metabolism in Table 4, and the predicted probabilities for the toxicological endpoints are presented in Table 5.

Data presented in Table 3 show quite distinct values for the predicted probabilities/activities. Both TTZ stereoisomers revealed high intestinal absorption and were not considered substrates and inhibitors of permeability glycoprotein. (R)-TTZ was not considered to be able to penetrate the blood-brain barrier and disclosed a moderate clearance and high plasma protein binding, limiting its partitioning from the blood into the tissues where it is metabolized. (S)-TTZ revealed an optimal distribution and a higher clearance but exposed a reasonable probability to penetrate the blood-brain barrier and, consequently, produce neurotoxicity. The possibility of TTZ penetrating the blood-brain barrier has already been noted in another computational study [18] that considered SwissADME [75] as a prediction tool based on models other than those used in the ADMETLab2.0 tool.

Data presented in Table 4 illustrate high probabilities for the two isomers to inhibit CYP1A2 and CYP2C19 and reasonable probabilities to be substrates for CY3A4. (R)-TTZ also exposed a high probability to inhibit CYP3A4 and a reasonable probability to inhibit CYP2C9. This outcome is in good agreement with another computational study [18] based on the SwissADME prediction tool that revealed the ability of TTZ to act as an inhibitor of CYP2C9 and CYP2C19. The interactions of the TTZ stereoisomers with human CYPs are addressed further using the molecular docking approach.

Both TTZ isomers revealed high probabilities of producing cardiotoxicity by the blockage of the h-ERG potassium channel. Furthermore, (R)-TTZ revealed a reasonable probability of producing skin sensitization, and (S)-TTZ emphasizes a high probability of producing carcinogenicity and respiratory toxicity. The skin sensitization potential and carcinogenicity of TTZ have already been predicted [18] using the PredSkin [76] and, respectively, CarcinoPred-El [77] computational tools.

It must be emphasized that the main limiting point of the outcomes predicting various biological effects is the reliability of the predictions, as it is widely known that computational models have limited domains of applicability (AD). The AD is defined as the range of the physicochemical, structural, or biological space where the model is expected to be exploited and the predictions are considered to be trustable [78]. In the case of the ADMETLab 2.0 prediction tool, the AD approach allows for estimating the prediction accuracy for each compound individually and making discriminating predictions with a precision close to that of the experimental data used for building the models. In the particular case of the TTZ stereoisomers, the predictions were within the applicability domain of the prediction tool.

The predictions obtained using the ENDOCRINE DISRUPTOME [43] tool regarding the effects of the TTZ enantiomers on the human nuclear receptors are revealed in Table 6. Data presented in this table show that TTZ enantiomers may have a low effect on estrogen α and β and the glucocorticoid, mineralocorticoid, and thyroid β receptors. (R)-TTZ had a low antagonistic effect and (S)-TTZ had a moderate antagonistic effect on the androgen receptor. The low endocrine disruption potential of TTZ was observed previously [18].

### 3.5. Molecular Docking Study Regarding the Interactions of the Enantiomers of Triticonazole with the Human Cytochromes and Plasma Proteins

A molecular docking study was implemented in order to assess the predictions obtained using the ADMETLab2.0 software regarding the ability of the TTZ enantiomers to interact with plasma proteins and human cytochromes and the results are presented in Table 7. Each chain of the multimeric proteins was considered for molecular docking to take into account the flexibility of the active site.

The two TTZ stereoisomers were able to bind to the plasma proteins, (S)-TTZ revealing higher interaction energies with both AGP and HSA. This outcome is in good correlation with the prediction obtained using ADMETLab2.0 and reveals the optimal distribution for both stereoisomers. With the exception of CYP2D6, the TTZ stereoisomers were similarly able to bind to the active sites of the human CYPs. Stronger interactions were revealed by the two TTZ stereoisomers with CYP1A2, which was also in good correlation with the highest values of the probabilities computed using ADMETLab2.0 for the two isomers to inhibit CYP1A2. Usually, the (S)-TTZ emphasizes higher interacting energies with human CYPs than (R)-TTZ.

The outcomes obtained using PLIP software [74] regarding the identified interactions between the two TTZ stereoisomers and the human plasma proteins and cytochromes are presented in Appendix A. For comparison purposes, PLIP software was also used for identifying the interactions appearing in the complexes of these proteins and the ligands that were present in the crystallographic structures and these data are also presented in Appendix A. The noncovalent interactions of the TTZ stereoisomers with the plasma proteins and human cytochromes were mainly hydrophobic, which was in correlation with the hydrophobic character of TTZ and hydrogen bonds. From one isomer to another, the spectra of the interactions made by TTZ stereoisomers with the residues of the investigated proteins were quite different, highlighting the specificity of the binding of each stereoisomer to these proteins and, therefore, the distinct biological activity.

It should be noted that the results obtained using the molecular docking approach were significant for the two TTZ stereoisomers, which highlights that incorporating stereochemically aware descriptors increases the accuracy of the predictive models used in ADMET screening.

Our study revealed the distinct biological activities of the two stereoisomers of TTZ. Specific literature has also emphasized cases regarding the differences in the toxicological and environmental properties of stereoisomers, but these distinct properties are not usually considered by regulatory agencies. At present, few regulations focus on the dietary and environmental hazards of chiral compounds. The most common approaches to assessing the risks overlook the implications of chirality on the biological activity of chemical compounds and this weakens the accuracy of the risk assessments [79]. However, taking into account the considerable number of plant protection products being marketed as a mixture of stereoisomers or that use metabolites with stereoisomers, the research interest in the implications of chirality on the ADMET properties of chemicals has grown significantly in the past few decades. Current EU pesticide regulations establish that stereoisomers should be treated as different chemical compounds for risk assessments and provide guidance regarding the use of products containing stereoisomers, supported by available scientific information [80]. In this context, the computational assessment of the toxicity of stereoisomers is a useful tool, especially in situations when the information regarding the distinct isomers is not available or is difficult to obtain, as it can guide experiments and reduce animal testing. The outcomes of this study also emphasize the necessity of considering chirality in risk assessment regulations.

## 4. Conclusions

Data presented in this study revealed that the application of the fungicide triticonazole on barley and wheat seeds sown in chernozem soil in the 14 days after being sown, caused a 70% decrease in the DHA and a 40% decrease in the PhA for both types of seeds, an approximately 20% increase in the UA in the case of the barley seeds, and a 40% increase in the UA in the case of the wheat seeds. This is consistent with the decrease in the total number of soil microorganisms after 14 days of incubation. This highlights that the fungicide may affect non-target organisms as soil microorganisms are linked functionally and/or nutritionally and any alteration in a component influences the entire community. The observed effects on the community of microorganisms and activity of enzymes found in soil were usually not dose dependent.

Regarding the effects on human health, both (R)- and (S)-TTZ revealed several biological effects. They were able to bind to plasma proteins being easily distributed through the human body, inhibit the human CYPs responsible for the metabolism of xenobiotics, reveal cardiotoxicity by the blockage of the h-ERG potassium channel, and affect the estrogen, glucocorticoid, mineralocorticoid, and thyroid β receptors. Several distinct pharmacokinetics and toxicological effects were identified for each of the TTZ stereoisomers. (R)-TTZ could produce skin sensitization and have a low antagonistic effect on the androgen receptor. (S)-TTZ could produce carcinogenicity and respiratory toxicity and have a moderate antagonistic effect on the androgen receptor. Furthermore, (S)-TTZ usually showed higher interaction energies both with plasma proteins and with human CYPs.

The limiting points of the outcomes of our study are common to laboratory conditions and in silico predictions. The laboratory experiments were performed in an artificial environment and are valid only for the specific created conditions (type of soil, low temperature, humidity variations, etc.). The limitation of the in silico studies is represented by the reliability of the predictions. The predictions were within the applicability domains of the considered models, and this increased the trustworthiness of the results. This study also emphasized that molecules’ isomeric structural characteristics should be considered when the models are built for in silico studies. Another limitation of the computational study is the fact that the predictions did not take into account the concentrations of the investigated molecules.

The potential importance of the results of the present study is that it highlighted the possible stereoselective effects of triticonazole both on human health and the soil environment. Furthermore, the outcomes of the study emphasize that human and environmental toxicity of (R)-TTZ is lower than that of (S)-TTZ, being in good agreement with other published data revealing higher acute toxicity of (S)-TTZ to aquatic and terrestrial non-target organisms compared to that of (R)-TTZ and that of the racemic mixture [7]. The obtained predictions can be included in tests with suitable designs that are meant to understand the role of stereoselectivity in the human and environmental toxicity of triticonazole. Taking into account that among the agrochemicals, more than 30% are chiral compounds [81], stereoselectivity should be considered in the safety assessments and regulatory decisions. This underlines that proper agricultural management should consider only (R)-TTZ instead of the racemate. Being known for the higher bioactivity of (R)-TTZ against fungi [5,6], the use of this stereoisomer also allows for the reduction of the quantity of fungicide used to benefit food safety and environmental protection.

## Figures and Tables

**Figure 1 molecules-27-06554-f001:**
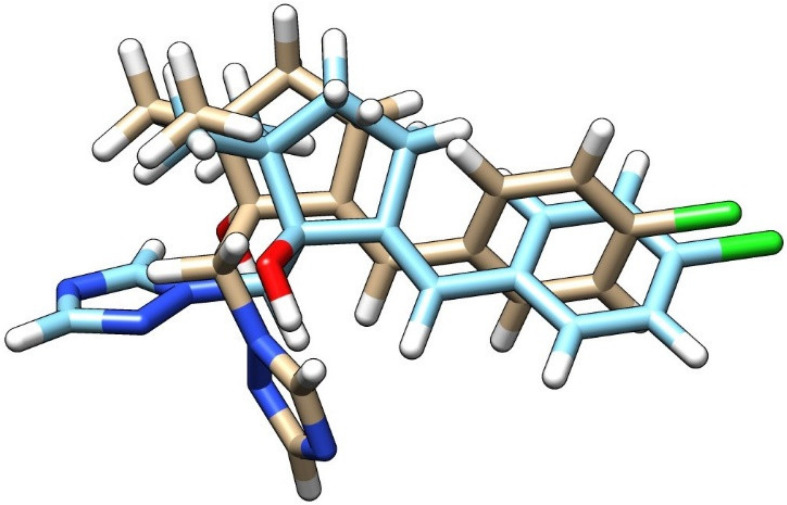
Chemical structures of the stereoisomers of triticonazole: (R)-triticonazole (carbon—brown), (S)-triticonazole (carbon—light blue), nitrogen—blue, oxygen—red, chloride—green, hydrogen—white.

**Figure 2 molecules-27-06554-f002:**
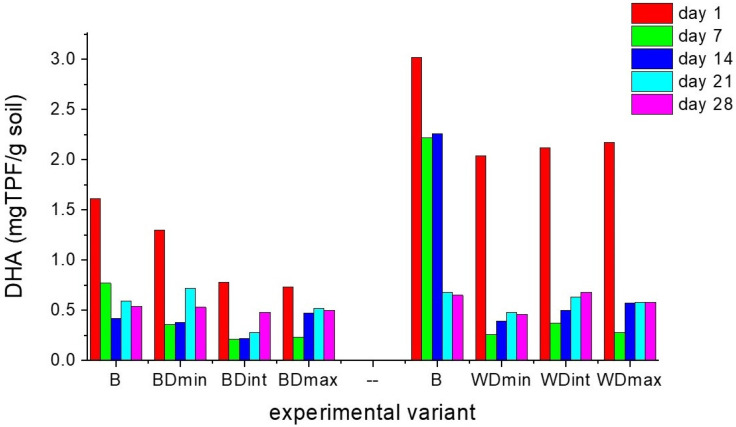
Soil dehydrogenase activity (DHA) for each experimental variant, determined every 7 days for a period of 28 days: B—seeds untreated with triticonazole, BDmin, and WDmin—barley and wheat seeds treated with the minimum dose, DBint, and WDint—barley and wheat seeds treated with the intermediate dose, BDmax, and WDmax—barley and wheat seeds treated with the maximum dose of triticonazole.

**Figure 3 molecules-27-06554-f003:**
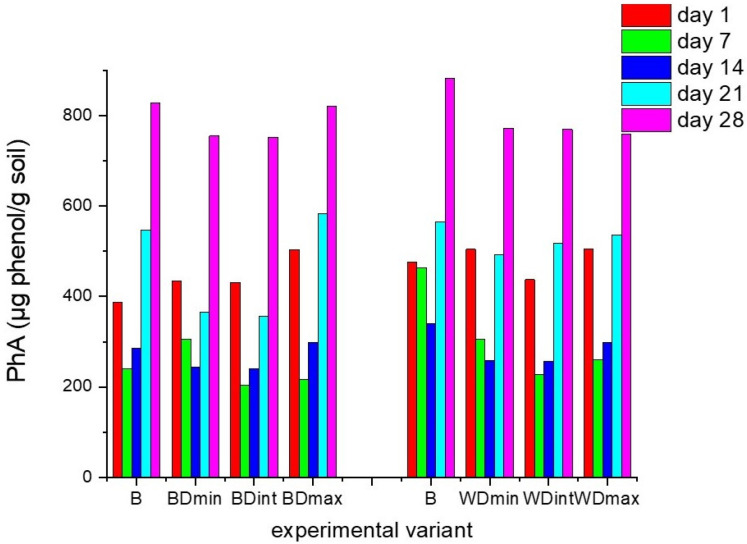
Soil phosphatase activity (PhA) for each experimental variant, determined every 7 days for 28 days: B—seeds untreated with triticonazole, BDmin, and WDmin—barley and wheat seeds treated with the minimum dose, DBint, and WDint—barley and wheat seeds treated with the intermediate dose, BDmax, and WDmax—barley and wheat seeds treated with the maximum dose of triticonazole.

**Figure 4 molecules-27-06554-f004:**
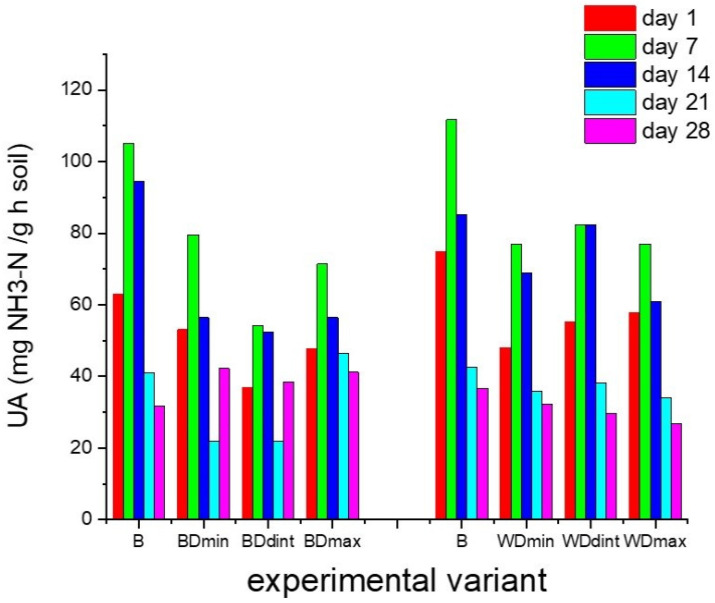
Soil urease activity (UA) for each experimental variant, determined every 7 days for 28 days: B—seeds untreated with triticonazole, BDmin, and WDmin—barley and wheat seeds treated with the minimum dose, DBint, and WDint—barley and wheat seeds treated with the intermediate dose, BDmax, and WDmax—barley and wheat seeds treated with the maximum dose of triticonazole.

**Figure 5 molecules-27-06554-f005:**
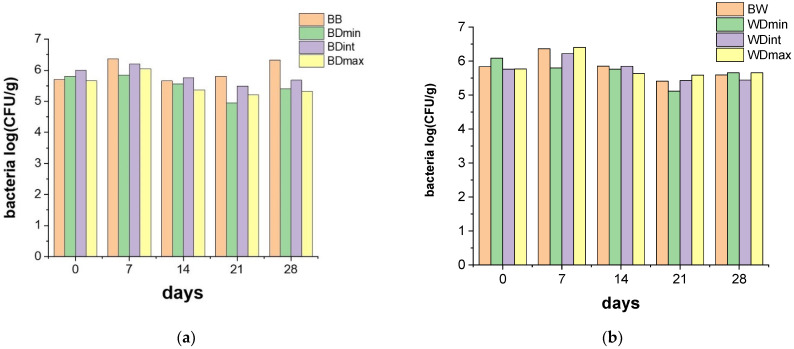
The effects of the treatment of barley (**a**) and wheat seeds (**b**) with TTZ on the communities of bacteria, recorded every 7 days for 28 days: BB and BW—seeds untreated with triticonazole, BDmin, and WDmin—barley and wheat seeds treated with the minimum dose, DBint, and WDint—barley and wheat seeds treated with the intermediate dose, BDmax, and WDmax—barley and wheat seeds treated with the maximum dose of triticonazole.

**Figure 6 molecules-27-06554-f006:**
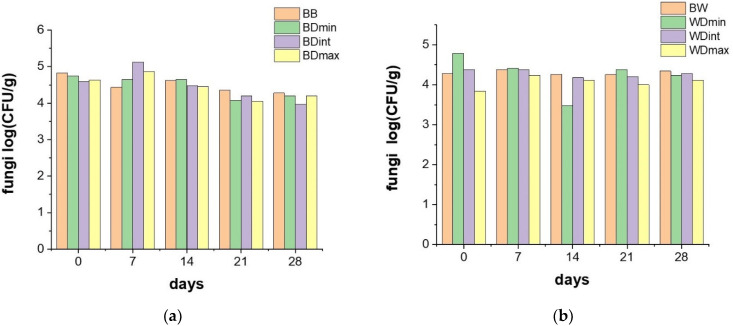
The effects of the treatment of barley (**a**) and wheat seeds (**b**) with TTZ on communities of fungi, recorded every 7 days for 28 days: BB and BW—seeds untreated with triticonazole, BDmin, and WDmin—barley and wheat seeds treated with the minimum dose, DBint, and WDint—barley and wheat seeds treated with the intermediate dose, BDmax, and WDmax—barley and wheat seeds treated with the maximum dose of triticonazole. The population of fungi in the soil cultivated with wheat and barley seeds treated with TTZ decreased slightly during the first 21 days of incubation; the decrease was not dependent on the dose of TTZ used (Figure 6).

**Figure 7 molecules-27-06554-f007:**
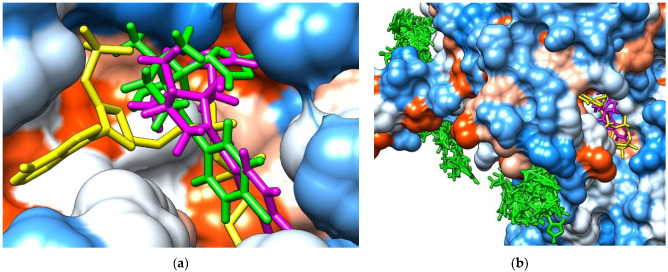
The binding modes corresponding to the highest interaction energies recorded for the binding of the (R)-TTZ and (S)-TTZ to the *Clostridium beijerinckii* NADP-dependent alcohol dehydrogenase (**a**) and the aldehyde dehydrogenase from *Bacillus cereus* (**b**). The proteins are shown as hydrophobicity surface (blue regions are hydrophilic, orange regions are hydrophobic) the NADPH is shown as yellow sticks, (R)-TTZ as magenta sticks, and (S)-TTZ as green sticks.

**Table 1 molecules-27-06554-t001:** Experimental variants considered in this study.

Experimental Variant/Dose	Seeds Untreated with Triticonazole	Minimum Dose 0.3 L/1000 kg	Intermediate Dose 0.6 L/1000 kg	Maximum Dose 0.9 L/1000 kg
Soil sown with seeds of barley	BB	BD_min_	BD_int_	BD_max_
Soil sown with seeds of wheat	BW	WD_min_	WD_int_	WD_max_

Soil samples were incubated under laboratory conditions for 28 days.

**Table 3 molecules-27-06554-t003:** Predictions regarding the absorption, distribution, and excretion of the stereoisomers of triticonazole (TTZ) in the human organism: HIA—human intestinal absorption, P-gp—permeability glycoprotein, BBB—blood–brain barrier, PPB –plasma protein binding, CL—clearance.

Stereoisomer	HIA < 30%	Pgp Substrate	Pgp Inhibitor	BBB Permeation	PPB	CL (mL/min/kg)
(R)-TTZ	0.004	0.008	0.056	0.333	94.66%	8.518
(S)-TTZ	0.003	0.008	0.011	0.514	89.39%	10.572

**Table 4 molecules-27-06554-t004:** Predictions regarding the metabolism of the stereoisomers of triticonazole (TTZ): CYP—cytochrome, s—substrate, i—inhibitor.

Stereoisomer	CYP 1A2 s	CYP 1A2 i	CYP 2C19 s	CYP 2C19 i	CYP 2C9 s	CYP 2C9 i	CYP 2D6 s	CYP2 CD6 i	CYP 3A4 s	CYP 3A4 i
(R)-TTZ	0.392	0.975	0.452	0.922	0.281	0.795	0.122	0.516	0.816	0.873
(S)-TTZ	0.481	0.976	0.484	0.894	0.104	0.564	0.066	0.414	0.718	0.790

**Table 5 molecules-27-06554-t005:** Predicted toxicological endpoints of the stereoisomers of triticonazole (TTZ): hERG—cardiotoxicity, HT—hepatotoxicity.

Stereoisomer	hERG	HT	AMES Mutagenicity	Skin Sensitization	Carcinogenicity	Eye Corrosion	Eye Irritation	Respiratory Toxicity
(R)-TTZ	0.878	0.412	0.488	0.779	0.641	0.004	0.273	0.273
(S)-TTZ	0.894	0.522	0.232	0.635	0.882	0.004	0.165	0.796

**Table 6 molecules-27-06554-t006:** Predictions regarding the inhibition of the nuclear receptors by the triticonazole (TTZ) stereoisomers: green cells—no effect, yellow cells—low effect, orange cells—moderate effect.

Nuclear Receptor	(R)-TTZ	(S)-TTZ
AR		
AR an		
ER α		
ER α an		
ER β		
ER β an		
GR		
GR an		
LXR α		
LXR β		
MR		
PPAR α		
PPAR β		
PPAR γ		
PR		
RXR α		
TR α		
TR β		

**Table 7 molecules-27-06554-t007:** Interaction energies (ΔG) between the stereoisomers of triticonazole (TTZ) and human plasma proteins and cytochromes, respectively. This table also contains the interaction energies obtained for the docking of the ligands that were present in the crystallographic structures of the investigated enzymes: AGP—alpha-1 acid glycoprotein, HSA—human serum albumin, CYP—cytochrome.

Human Protein	PDB ID	Ligand in Crystallographic Structure		(R)-TTZ		(S)-TTZ	
		ΔG (kcal/mol)	Mean Value of ΔG (kcal/mol)	ΔG (kcal/mol)	Mean Value of ΔG (kcal/mol)	ΔG (kcal/mol)	Mean Value of ΔG (kcal/mol)
Plasma protein	AGP	−6.14	−6.14	−7.36	−7.36	−7.78	−7.78
	HSA chain A	−7.37	−7.44	−9.25	−8.86	−8.52	−9.37
	HSA chain I	−7.50		−8.47		−10.21	
Cytochrome	CYP1A2 chain A	−8.62	−8,62	−8.51	−8.51	−8.83	−8.83
	CYP2C9 chain A	−7.73	−7.92	−7.78	−7.50	−7.32	7.60
	CYP2C9 chain B	−8.11		−7.22		−7.88	
	CYP2C19 chain A	−7.47	−7.50	−7.35	−7.24	−7.38	−7.49
	CYP2C19 chain B	−7.40		−7.16		−8.73	
	CYP2C19 chain C	−7.63		−6.96		−6.52	
	CYP2C19 chain D	−7.51		−7.47		−7.34	
	CYP2D6 chain A	−8.71	−8.71	It did not bind in the active site			
	CYP3A4 chain A	−6.67	−6.67	−8.97	−8.79	−7.63	−7.63

## Data Availability

Not applicable.

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
