# Peer review of "Assessment of the Effects of Triticonazole on Soil and Human Health"

_molecules, 2022, doi:10.3390/molecules27196554_

Round 1

Reviewer 1 Report

The authors did a study titled "Assessment of the effects of triticonazole on soil and human health." The study is generally in good shape, well-organized, and within the scope of the journal. The data can assist local regulatory agencies in managing and regulating pesticide residues in the soil in order to minimize public health risks. There are some suggestions for the authors to improve the quality of the paper.

The authors are able to determine if the structure of the research article published in the journal is appropriate. As is typical, the method section should precede the result part.

The authors should elaborate on the regulatory implications of their findings.

Author Response

Thank you for your observations and suggestions, they are clearly meant to improve the quality of the manuscript. We have considered all of them and in the following we illustrate how they are implemented in the manuscript. Your text is in italics and our answer is in normal text.

The authors did a study titled "Assessment of the effects of triticonazole on soil and human health." The study is generally in good shape, well-organized, and within the scope of the journal. The data can assist local regulatory agencies in managing and regulating pesticide residues in the soil in order to minimize public health risks. There are some suggestions for the authors to improve the quality of the paper. The authors are able to determine if the structure of the research article published in the journal is appropriate. As is typical, the method section should precede the result part.

In the original manuscript we put the Methodology section as it was asked in the Instruction for Authors. Now the Methodology section precedes the Results and Discussion section. All the references were renumbered accordingly.

 The authors should elaborate on the regulatory implications of their findings.

New text has been added in order to emphasize the regulatory implications, lines 547-565.

Reviewer 2 Report

Thank you for the invitation to review the article titled “Assessment of the effects of triticonazole on the soil and on the human health”. The authors have not only determined the ecological effects of TTZ on the enzyme activity in soil but also evaluated the effects of TTZ on humans. Multiple methods, e.g., experimental and in silico methods, have been successfully applied in this study. The results showed soil microorganisms will be greatly impacted by TTZ exposure and TTZ could induce skin irritation, carcinogenicity, and respiratory toxicity. However, I have not seen a connection between the ecological effects and human health of TTZ across the manuscript. How will the human be exposed to these chemicals, e.g., exposure pathways? I think it important for the authors to address this concern at least in the section of the Introduction. In addition, could the authors please introduce why barley and wheat seeds were selected and treated with TTZ? I believe there are other plant species that were sensitive to environmental pollutants. Other than these two points, this manuscript is well-written and organized. I recommend the publication of this manuscript after my two major concerns and the following minor questions are answered.

Minor comments:

1.     Line 57: the half-life of TTZ in soil was 36.1-242 days, can we say it is persistent if the half-life is below 180 days

2.     Figure 1: the values of DHA for barely and wheat seeds are quite different. Could the authors please explain it?

3.     The authors should spell the acronyms of BD, B, and WD in the captions in the figures.

4.     Table 7: are these levels environmentally relevant?

Author Response

Thank you for your observations and suggestions, they are clearly meant to improve the quality of the manuscript. We have considered all of them and in the following we illustrate how they are implemented in the manuscript. Your text is in italics and our answer is in normal text.

Thank you for the invitation to review the article titled “Assessment of the effects of triticonazole on the soil and on the human health”. The authors have not only determined the ecological effects of TTZ on the enzyme activity in soil but also evaluated the effects of TTZ on humans. Multiple methods, e.g., experimental and in silico methods, have been successfully applied in this study. The results showed soil microorganisms will be greatly impacted by TTZ exposure and TTZ could induce skin irritation, carcinogenicity, and respiratory toxicity. However, I have not seen a connection between the ecological effects and human health of TTZ across the manuscript. How will the human be exposed to these chemicals, e.g., exposure pathways? I think it important for the authors to address this concern at least in the section of the Introduction.

In the Introduction section, we have introduced several sentences and reorganized the text such as to reveal the connection between the ecological effects and the human health and emphasized the pathways of the human exposure to TTZ (lines 67-75).

In addition, could the authors please introduce why barley and wheat seeds were selected and treated with TTZ? I believe there are other plant species that were sensitive to environmental pollutants.

The Reviewer is right, triticonazole is used for treating seeds and also as foliar spray for numerous plat species, but it is used especially for treating cereals seeds. An argumentation of using wheat and barley seeds treated with triticonazole in our study is introduced in the text (lines 131-134).

Other than these two points, this manuscript is well-written and organized. I recommend the publication of this manuscript after my two major concerns and the following minor questions are answered.

 Minor comments: 

  1. Line 57: the half-life of TTZ in soil was 36.1-242 days, can we say it is persistent if the half-life is below 180 days

We have interpreted the half-life of TTZ in soil in accordance with AGRICULTURAL SUBSTANCES DATABASES -background and support information (http://sitem.herts.ac.uk/aeru/ppdb/en/docs/5_1.pdf). This document specifies for the degradation rates as following: < 30 = Non-persistent; 30 - 100 = Moderately persistent; 100 - 365 = Persistent; > 365 = Very persistent. We have corrected the text and added the 90% degradation time instead of the half-life (line 57).

  1. Figure 1: the values of DHA for barely and wheat seeds are quite different. Could the authors please explain it?

An explanation has been added in the main text, lines 273-283.

  1. The authors should spell the acronyms of BD, B, and WD in the captions in the figures.

The acronyms BD, B and WD have been added in the figures captions.

  1. Table 7: are these levels environmentally relevant?

The doses that were used in the present study correspond to the recommended by the producer as minimum and maximum dose for wheat and barley seeds. An intermediary dose has been also considered. This information is added in the manuscript, lines 134-136.
